

# Monitoring crustal $CO_2$ flow: methods and their applications to the mofettes in West Bohemia

Tomáš Fischer, Josef Vlček, Martin Lanzendörfer

Charles University, Faculty of Science, Prague, Czechia

**Abstract** Monitoring of $CO_2$ degassing in seismoactive areas allows the study of correlations of gas release and seismic activity. Reliable continuous monitoring of the gas flow rate in rough field conditions requires robust methods capable of measuring gas flow at different types of gas outlets such as wet mofettes, mineral springs and boreholes. In this paper we focus on the methods and results of the long-term monitoring of $CO_2$ degassing in the West Bohemia/Vogtland region in Central Europe, which is typified by the occurrence of earthquake swarms and emanations of carbon dioxide of magmatic origin. Besides direct flow measurement using flowmeters, we introduce a novel indirect technique based on quantifying the gas bubble contents in a water column, which is capable of functioning in severe environmental conditions. The method calculates the mean bubble fraction in a water-gas mixture from the pressure difference along a fixed depth interval in a water column. Laboratory tests indicate the nonlinear dependence of the bubble fraction on the flow rate, which is confirmed by empirical models found in the chemical and nuclear engineering literature. Application of the method in a pilot borehole shows a high correlation between the bubble fraction and measured gas flow rate. This was specifically the case of two coseismic anomalies in 2008 and 2014, when the flow rate rose during a seismic swarm to a multitude of the pre-seismic level for several months and was followed by a long-term flow rate decline. However, three more seismic swarms occurring in the same fault zone were not associated with any significant $CO_2$ flow anomaly. We surmise that this could be related to the slightly farther distance of the hypocenters of these swarms than the two ones which caused the coseismic $CO_2$ flow rise. Further long-term CO2-flow monitoring is required to verify the mutual influence of CO2 degassing and seismic activity in the area.




## 1. Introduction

Long-term monitoring of crustal fluids activity provides a unique opportunity to better understand the relationships among tectonic processes, seismic activity and migration of fluids in the Earth crust. Carbon-dioxide of deep origin represents a link between deep seated magmatic sources of $CO_2$, the fluid migration paths in the crust, which are controlled by the tectonic stress field, and the earth surface. The presented study is focused to monitoring of $CO_2$ degassing in the West Bohemia/Vogltland area, which is located in the western part of the Bohemian Massif (BM), the largest coherent surface exposure of basement rocks in central Europe. The western BM is hosting a junction of three tectonometamorphic units, Saxothuringian, Teplá-Barrandian and Moldanubian (Franke, 2000). It is intersected by two regional tectonic structures, the NE-SW trending Eger Rift (ER) and NNW-SSE trending Mariánské Lázně Fault (MLF) (Fig. 1).

The Tertiary ER is a 300 km long striking structure characterized by elevated heat flow and Cenozoic volcanism and its formation is thought to be related to Alpine collision (Ziegler, 1992). The Late-Variscan MLF was reactivated several times during the geological history up to Cenozoic when it participated in the formation of the Cheb Basin (CB). CB is typified by a blocky structural fabric due to a network of faults. Besides the NNW and NW morphologically expressed marginal faults also faults striking NE, E-W and N-S were identified within the basin (Špičáková et al., 2000; Bankwitz et al., 2003).

The present geodynamic activity is manifested by earthquake swarms, massive $CO_2$ degassing of mantle origin and Quaternary volcanism (Fischer et al., 2014). Seismic activity in the form of earthquake swarms is concentrated in the area of CB, in particular the Nový Kostel focal zone (Fig. 1), where more than 80% of seismic energy is released in frames of the whole seismogenic region. Here the hypocenters form a N-S trending, steeply dipping belt in the depth range from 6 to 10 km; however, no clear fault outcrop has been identified that would match the focal zone geometry. The prevailing focal mechanisms coincide very well with the orientation of the fault zone of 169°. Inversion of focal mechanisms for stress field yields maximum compression direction in the range N135–155°E, which coincides well with the average direction N144°E in Western Europe (Fischer et al., 2014). This direction is however parallel to the strike of the MLF, which indicates a passive role of the MLF in the present stress field (Vavryčuk, 2011).

The strongest earthquakes usually do not exceed $M_L$ 4.5, as was the case of all the eight major instrumentally recorded swarms between 1985 and 2018.

The concentration of the geodynamic phenomena in this small region is not clearly understood. Some authors relate this seismic activity to intersecting crustal faults (e.g., Bankwitz et al., 2003) or to fluids of mantle origin (e.g., Bräuer et al., 2003; Babuška et al., 2016), which could originate from active magmatic underplating (Hrubcová et al., 2017). The degassing occurs in the form of $CO_2$-rich mineral waters and wet and dry mofettes in several degassing fields. Carbon dioxide is the carrier phase for mantle-derived minor components such as helium, the isotope ratios of which are the best tool to determine whether the fluids are of crustal or mantle-derived origin; high $^3He/^4He$ ratios indicate that ascending gases are of mantle origin (Bräuer *et al.*, 2003).

Gas flow is concentrated in three degassing centers: Cheb Basin, Mariánské Lázně, and Karlovy Vary (KV) (Weinlich *et al.*, 1999; Geissler *et al.*, 2005; Kämpf *et al.*, 2007). They are characterized by a high gas flow with $CO_2$ concentrations of more than 99 vol. %. Cheb Basin also has the highest concentration of seismic activity, which makes





it ideal for studying the relations between seismicity and gas flow. Interestingly, numerous studies of the local
earthquake swarms show that they may be related to pressurized fluid in the crust and the ascent of gas. This has been
pointed out by numerous researchers including Špičák and Horálek (2000), Hainzl and Fischer (2002), Fischer and
Horálek (2005) and Hainzl et al. (2016), based on space-time analysis of the seismicity, Horálek et al. (2002),
Vavryčuk (2002) and Vavryčuk and Hrubcová (2017), on the basis of moment tensor analysis, and Dahm and Fischer
(2014), and Bachura and Fischer (2016), based on Vp/Vs analysis of the volume of hypocenters. The last two studies
show that compressible fluids are required to explain the low velocity ratio observed in the course of seismic activity.
The gases produced in the West Bohemia/Vogtland mineral springs and mofettes show high $^3$He/$^4$He ratios; these are
significantly higher than the average continental crust, indicating their mantle-derived origin. Also, the $\delta^{13}$C values in
the $CO_2$-rich gas escapes indicate their origin in the upper mantle (Weinlich *et al.*, 1999; Bräuer *et al.*, 2003). The
highest portions of mantle-derived helium (up to 6 $R_a$, where $R_a$ corresponds to the $^3$He/$^4$He ratio of the atmosphere)
were found in the CB; the KV degassing center has the lowest $^3$He/$^4$He ratios of 2.5 $R_a$. Lower He-isotope ratios (e.g.
$^3$He/$^4$He < 6Ra) probably reflect the gas mixing with crustal-derived He along fluid pathways (Bräuer *et al.*, 2008).
$CO_2$ flow monitoring in West Bohemia has been conducted since the 1990s in a rather discontinuous way. The longest-
running observation project is probably the monitoring of Radon activity in Bad Brambach (Heinicke and Koch, 2000;
Koch et al., 2003), which has been conducted since 1989.
Another long-time monitoring was carried out as part of the "Research of $CO_2$ pressure field in the area of West
Bohemian spas" project funded by the Ministry of the Environment of the Czech Republic from 1996 to 2005. Gas
flow in open boreholes and gas pressure in closed boreholes was monitored at 11 gas escape sites in the Cheb Basin
and near Mariánské Lázně (Škuthan et al., 2001, Hron et al., 2006). Monitoring of pressure in a closed well was
preferred at many project sites since the functioning of mechanical flowmeters was unreliable due to condensation
and freezing. A different type of $CO_2$ flow monitoring was carried out by Faber et al. (2009), who measured diffuse
gas flow by determining $CO_2$ concentration in soil gas at two stations in the Nový Kostel fault zone. $CO_2$ flow
monitoring was also conducted by Heinicke (personal communication) in the Bublák mofette from 2008 to 2014 by
recording the acoustic noise of bubbles below the water table, a method which is similar to that used by Koch et al.
(2003). No convincing observation of seismogenic $CO_2$ flow anomaly was, however, presented in the above-
mentioned studies.
Mapping of $CO_2$ emanations was conducted in the area by Nickschick et al. (2015). They used an infrared gas analyzer
and accumulation chambers to measure $CO_2$ flux and $CO_2$ soil concentration in the mofette field of Hartoušov and
found that the diffuse gas flow in dry vents accounts for a high portion of the mofette field's total gas production.
The measurement of $CO_2$ flow presented in this paper began in 2009 in the Hartoušov mofette field with the use of a
laboratory chamber flowmeter. Despite problems from the condensation of moisture and freezing temperatures, which
resulted in time series gaps, we observed a massive post-seismic $CO_2$ flow increase shortly after the first $M_L$ 3.5
mainshock of the 2014 seismic sequence. A comparison with the fault valve model showed a striking fit, which
indicated that the earthquake fracture released gas accumulated in the reservoir beneath hypocenters (Fischer et al.,
2017). This gave us a reason to extend the monitoring and test different, more durable, gas flow measurement methods.
In this paper, we introduce the principles for our approaches and give a basic comparison of them. We also present





the data recorded from the Hartoušov, Bublák, Soos and Prameny stations and evaluate their response to air pressure
and temperature and their possible relation to seismicity.

**2. Data and methods**
Two types of $CO_2$ degassing are observed in West Bohemia/Vogtland: (i) diffuse gas flow in soil and (ii) massive gas
production in mofettes and mineral springs. While gas diffusion in soil is influenced by soil moisture and other local
conditions, among other factors, gas flow in massive sources is independent of environmental conditions and should
reflect the influence of the gas source at in the depth. The deep roots of $CO_2$ mofettes were also documented by a
massive increase in $CO_2$ flow in the Hartoušov mofette that began about four days after the start of seismic activity in
2008 and 2014 (Fischer et al., 2017). This points to the relatively fast speed of gas migration in the upper crust and
qualifies mofettes as favorable places to monitor the amount of leaking gas. Since 2015, the current monitoring at
Hartoušov has been extended to other places in order to provide robust measurements capable of recording possible
future gas anomalies at multiple sites. Because the conditions differed among the monitored sites, different
measurement methods were designed. In this study, we distinguish between *direct* and *indirect* gas flow measurement
methods (Camarda et al. 2006). The direct methods directly record the volume of gas per minute and require that gas
flow be captured by a funnel or borehole. The indirect methods either involve deriving gas flow from the bubble
fraction in water (pressure probes are placed beneath the water table), or rely on measuring gas overpressure in a
closed borehole or, finally, they calculate $CO_2$ flux from the concentration of gradients in the soil (Baubron et al.,
1990). The dynamic concentration method is based on measuring the $CO_2$ content in a mixture of soil gas and air
obtained by a special probe placed vertically in the soil. The dynamic concentration is proportional to the soil CO2
flux according to an empirical relationship, which depends on soil permeability (Gurrieri and Valenza, 1988).

*Figure 1 .*
**2.1 Monitoring network**
Five gas escape sites were monitored in the period described: *Hartoušov, Bublák, Soos, Dolní Částkov* and *Prameny*
(see the map in Fig. 1 and photos in Fig. 2). While the first three are located in mofette fields, the remaining are
boreholes which tap mineral spring sources.
The pilot site of *Hartoušov* is located in a wooden hut above a 28.2 m deep borehole, which taps a $CO_2$ saturated,
pressurized aquifer. The plastic borehole casing, with an inner diameter of 115 mm, is perforated in the depth range
of 20-28 m. Water level measurements date back to 2007, and gas flow has been measured here using a drum chamber
gas flowmeter since 2009. The sensitivity of this type of instrument to environmental conditions (freezing or
evaporation of the working liquid) caused gaps in the recorded time series. Since 2013, there have only been brief
gaps thanks to the use of a different type of working liquid, improvements in the condenser separation and thermal
insulation. This direct field gas flow measurement is used as a reference for testing different flow measurement devices





prior to their installation at other sites. Additional permanent measurements include water pressure in several depth
levels, water temperature and air temperature and pressure.
The *Bublák* station has been located in a natural mofette in a swamp since 2015. To avoid interfering with natural
conditions at this site, the equipment is buried underground, which does not allow for direct gas flow measurement.
Instead, the differential water level is measured and used as a proxy for the volumetric fraction of free gas in water,
see section 2.3. Because of the rising bubbles, the water does not freeze in winter, making this measurement quite
stable. The *Soos* station has been located in a natural mofette field since 2015, and the gas from a single mofette is
captured by a funnel allowing for direct gas flow measurement. The small size of the metal box shelter and the need
to use battery power, however, do not make it possible to prevent the freezing of the system in winter. The water level
and temperature in the mofette and the volumetric fraction of free gas are measured here using an electric resistivity
probe. In *Dolní Částkov*, the gas escapes both through a shallow borehole and the surrounding soil, which makes the
flow measurements rather unstable. The *Prameny* station is located on top of a 100 m deep closed borehole (HJ-3A,
drilled 1994) with degassing mineral water. Conditions at this site allow only for the measurement of the water level,
temperature and wellhead gas pressure, which have been available since 2009.
*Figure 2 .*
Table 1. CarbonNet monitoring network.

| Station name and code | Environment | Methods |
|---|---|---|
| *Bublák* <br> *BUB* | *Natural mofette* | *Water temperature, two pressure heads (sensor depths 0.7 and 1.4 m)* |
| *Hartoušov* <br> *HAR* | *30m deep open borehole VP8303* | *Air temperature, barometric pressure, three pressure heads (sensor depths 4.45, 5.45 and 27.2 m), water temperature, gas flow rate, differential pressure in the well* |
| | *105.8m deep closed borehole* | *Pressure head (sensor depth 92m), water temperature, absolute wellhead pressure* |
| *Prameny* <br> *PRA* | *100m deep open borehole HJ-3A* | *Pressure head (sensor depth 4.5 m), water temperature, absolute wellhead pressure* |
| *Soos* <br> *SOO* | *Natural mofette* | *Pressure head (sensor depth 1.5 m), water temperature, water resistivity* |
| *Dolní        Částkov* <br> *DCA* | *10 m deep open borehole* | *Gas flow rate* |






## 2.2 Direct CO₂ flow measurement methods

Long-term gas flow monitoring in the field must meet various requirements. It should provide sufficiently accurate
data of gas flow, which may contain dirt particles and moisture in changing field conditions of temperature, humidity
and air pressure. The presence of carbon dioxide further creates a highly corrosive environment, which the sensors
should withstand. Commercial flowmeters are usually not designed to meet these demands. We have tested (at SOOS
and Dolní Částkov stations) the *MEMS* (Micro-Electro-Mechanical Systems) *flowmeter*, which is based on heat
convection in moving gas. It works on the principle of a wheatstone bridge, where changes in the resistivity of the
resistor are measured according to the temperature changes caused by the flow of gas through a heater placed in the
middle of the sensor (Dmytriw et al., 2007). These low-cost sensors, however, failed in our tests. None of the MEMS
flowmeters tested measured for longer than 4 months, despite the installation of filters to capture solid particles and
moisture from the gas before entering the sensor. A popular way of measuring gas flow is the *Venturi type flowmeter,*
which works by measuring the drop-in pressure at a constriction in a tube. Our tests of similar devices failed due to
temperature drifts of the sensor and electronics, which were of the same order as the CO₂ flow variations. Direct flow
measurement methods also include the *acoustic method* based on the Doppler effect, which is commonly used for
water flow measurement. This, however, does not appear to be suitable for gas, which contains fewer particles acting
as diffractors than liquid.
The standard *flowmeters with rotating mechanical parts* driven directly by the gas flow were also found not suitable
due to the corrosive CO₂ environment. Better performance was achieved with a *drum-type chamber flowmeter*, which
contains a revolving measuring drum within a packing liquid (we use low-viscosity oil). The measuring drum
compulsorily measures volume by periodically filling and emptying four rigid measuring chambers. This *chamber*
*laboratory instrument* was found suitable for field measurement, where sufficient space and non-freezing temperatures
can be guaranteed. It has been used as the primary flowmeter at the Hartoušov station.

## 2.3 Indirect CO₂ flow measurement methods

### Gas pressure in a closed borehole

In a closed borehole tapping a gas-saturated aquifer an overpressure builds up whose magnitude has been speculated
to reflect the amount of gas entering the aquifer from below (e.g. Hron and Škuthan, 2006). However, a profound
discussion of how exactly the deep CO2 leakage affects the measured overpressure is still absent, to the best of our
knowledge. Considering a CO2 flux $q = q_1 + q_2$ summing the flux through the ceiling of the aquifer in the vicinity
of the borehole, $q_1$ and the possible gas leakage through the borehole, $q_2$, and assuming simply that the borehole
overpressure $p$ is proportional to the both, then $p$ follows the equation
$$p = \frac{q}{K_1 + K_2},$$

where $K_1$ and $K_2$ are the permeability factors related to the ceiling of the aquifer and to the borehole sealing,
respectively. Hence the measured overpressure is proportional to the gas flux controlled by deep processes, but also





influenced by the permeability of the superficial layer as well as by any possible leaks through the wellhead. In
particular, any variation in sealing layer properties, caused e.g. by the actual weather conditions, is then directly
projected onto the pressure measured.
Accordingly, in spite of the easy implementation of the pressure measurements in a closed borehole, we used this
method only at the *Prameny* site, where technical and logistic conditions did not allow the installation and maintenance
of a flow measurement. Excessive influence of $K_1$ on the measured pressure can be suppressed by introducing a
controlled leakage in the wellhead, which ensures that $K_2$ is not small in comparison to $K_1$ as has been implemented
at *Prameny* station.

**Bubble fraction in water**
We have used the bubble fraction monitoring method since observing a striking coincidence between the gas flow rate
and groundwater level (see later in this paragraph) increase in the Hartoušov borehole during the 2014 seismic
sequence (Fischer et al., 2017). Within a few months after the beginning of the sequence, the gas flow rate in the
borehole increased fivefold and the measured water level by more than 1 m. Since then, both quantities have indicated
an overall gradual decrease back to their original levels.
Instead of the notion of groundwater level, adopted in (Fischer et al., 2017) and other works, we stick to the more
strictly defined terms *pressure head* and *hydraulic head* in the present paper, which is due a few explanatory
comments. Within a steady water column resting in a borehole (or a narrow mofette), the hydraulic head (defined as
the sum of the pressure head and elevation) is independent of the elevation and is referred to as the groundwater level,
as it coincides with the elevation of the free water surface observed in the borehole. This is why the exact elevation of
the actual placement of the pressure probes in the borehole is usually disregarded, and the term groundwater level is
used somewhat loosely without risk of any confusion. In the case of a continuous bubbly flow through the borehole,
however, hydraulic head is not a depth-independent quantity, but rather inevitably increases with elevation. An
intuitive explanation is that the mean density of the water - gas bubbles mixture is markedly lower than that of water
(this is, however, merely an approximation, see also section 2.5). Following this simple concept, the density of the
mixture would be (disregarding the density of the gas $CO_2$ as negligible)

$$\varrho(z) = (1 - \phi(z))\varrho_w, \tag{1}$$

where $\phi(z)$ denotes the volumetric fraction of bubbles in the water column profile at elevation $z$, and $\rho_w$ stands for
the mass density of the water in the well (say, clear water at a given constant temperature). Denoting by $\psi(z)$ the
pressure head, related to the actual pressure $p(z)$ through $p(z) = \rho_w g \psi(z) + b$ with $g$ being the gravitational
acceleration and $b$ the barometric pressure (Fig. 3a), and denoting by $h(z) = \psi(z) + z$ the hydraulic head, we assert
that the difference in the measured pressures, pressure head and hydraulic head equals

$$p(z_1) - p(z_2) = p_1 - p_2 = \int_{z_1}^{z_2} \rho(z) g \; dz$$

$$\Rightarrow \; \psi_1 - \psi_2 = \int_{z_1}^{z_2} (1 - \phi(z)) \; dz \tag{2}$$

$$\Rightarrow \; h_2 - h_1 = \int_{z_1}^{z_2} \phi(z) \; dz,$$





i.e. the hydraulic head *h(z)* measured in the borehole increases with elevation by a factor equal to the volumetric
fraction of the bubbles in the borehole profile. Here, for the sake of brevity, we abstract from the time dependence of
all quantities.
*Figure 3*

The mean bubble fraction within the measured section of the water column can thus be defined as
$$\overline{\phi_{12}} = \frac{h_2 - h_1}{z_2 - z_1} = 1 - \frac{\psi_1 - \psi_2}{z_2 - z_1} \tag{3}$$

As the ascending gas bubbles expand due to the decreasing pressure, both the volumetric flux of the gas and the bubble
fraction $\phi(z)$ increase correspondingly with elevation. In order to obtain a quantity independent of the depths of the
pressure probes, a further correction needs to be applied. A reasonable approximation can be obtained based on the
following simplification. We assume that the gas expands isothermally, so that its volumetric flux is inversely
proportional to pressure, and that the bubble fraction is approximately proportional to the volumetric flux, so that we
can write
$$\phi(z) = \phi_0 \frac{p_0}{p(z)}$$

where $\varphi_0$ represents the bubble fraction at the reference pressure $p_0$ (which we later set as 100 kPa). Further,
approximating the pressure profile between the two pressure probes by a linear function

$$p(z) = p_1 + \frac{z - z_1}{z_2 - z_1}(p_2 - p_1)$$


we obtain (by substituting $\phi(z)$ to (2) and integrating) the formula for $\varphi_0$, let us call this the projected bubble fraction,

$$\phi_0 = \overline{\phi_{12}} \frac{p_2 - p_1}{p_0 \ln(p_2/p_1)} \tag{4}$$


One should note that the quantity obtained here is subject to some uncertainty due to a number of simplifications, and
that it only gives the approximated volumetric fraction and not the gas flow rate itself (see section 2.5).
In the Hartoušov borehole, the pressure head had been measured until September 2018 by one pressure gauge in the
depth of 8 meters, well above the bubble entry point. In (Fischer et al., 2017), the corresponding hydraulic head was
referred to as the groundwater level. As proposed in the paper, we split its time variation into two parts: the variation
(a) of the hydraulic head $h_e(t)$ at the bottom of the bubble flow column and (b) of its increase through the column due
to the gas bubbles. The optimal solution to obtain data for (a), which was implemented in Hartoušov in late 2018, is
to measure the pressure head in a depth beneath the bubble entry point directly by a dedicated pressure probe (Fig 3b).
While direct measurement was unavailable, it was supposed that (a) is given only by the surrounding hydrogeologic
situation and is unaffected by the gas flow. A single measurement of the pressure at the bubble occurrence depth by
(Fischer et al., 2017), corrected by a continuous pressure head record from a nearby observation well in Hrzín 8 km
apart, which is not affected by the $CO_2$ gas flow, was used as $h_e(t)$. Note that $h_e(t)$ also describes the hydraulic head
in any depth beneath the occurrence of bubbles. While (Fischer et al., 2017) considered the possibility that the gas





exsolution depth varies with time, we argue here (see section 2.4) that the gas bubbles have to appear at the penetrated
section of the Hartoušov borehole. This allows us to determine the mean volumetric fraction of the bubbles using eq.
(3) with $h_1(t) = h_m(t)$ being the hydraulic head measured at the depth $d_m = 4$ m, and $h_2(t) = h_e(t)$ being the hydraulic
head measured in any depth below the bubble entry depth, which we suppose to be at the upper part of the penetrated
section at $d_e = 20.5$ m (Fig.3).
In Fig. 4 the record of $h_e(t)$ and $h_m(t)$ and the resulting projected bubble fraction $\phi_0(t)$ defined by eq. (4) is shown
for the whole period studied in Fischer et al. (2017). We refer to this method as the *integral method.*

*Figure 4*

The method presented above is applicable only in boreholes and narrow tube-like mofettes. The borehole should tap
the underground water, and there should exist a continuous column of gas bubble flow from certain depth to surface.
Also, independent measurement of the hydraulic head in the aquifer/reservoir beneath the bubble flow column should
be possible, either in the same well or, at least, in a nearby well free of gas flow. These conditions are not fulfilled in
natural mofettes, which are usually only less than 2 m deep and communicate with the surface water significantly. In
such cases, the difference of pressure heads along a fixed depth interval *within* the bubble flow column can be
measured and used to define the mean bubble fraction $\overline{\phi_{12}}(t)$ and the projected bubble fraction $\phi_0(t)$.
This *differential method* has been tested in the Hartoušov well and Bublák mofette stations since 2015 using two
analog water-level sensors attached at a 1-meter distance on a metal rod. The obvious disadvantage is that both
measurements (instead of just one) are subject to fluctuation due to the bubbly flow, and that the noise in the resulting
bubble fraction data is inversely proportional to the distance between the probes. To suppress the noise, an RC circuit
with a 100 s time constant is applied.
An alternative way of determining the bubble fraction is based on the electric resistivity measurement of the water-
bubble mixture. Unlike the pressure difference method, this method does not need to be focused on the vertical chain
of bubbles, but it can assess the fraction of bubbles in a 3D volume defined by the geometry of electrodes. For this
purpose, two water resistivities are measured by a special probe in the mofette: the reference resistivity of the water
free of bubbles $R_R$ and the resistivity of the water-bubble mixture $R_M$. The bubble fraction is then derived as

$$\phi(t) = 1 - c \, \frac{R_R(t)}{R_M(t)}$$

where $c$ is the geometric calibration constant. This type of measurement has been tested in the Soos mofette since

291    2015.


**2.4 Depth of gas bubbles appearance**
It is possible to speculate that the exsolution of the gas bubbles from the water with dissolved $CO_2$ takes place at a
certain depth in the borehole, while below that depth the pressure is sufficient to contain the $CO_2$ in the dissolved
phase. In this view, the exsolution depth $d_e$ could vary in time, as considered by Fischer et al. (2017), following



variations of $h_e(t)$ and of the $CO_2$ supply from the reservoir. Let us note, however, that such a scenario is only possible
for gas fluxes much lower than those observed in the Hartoušov borehole.
Assuming a steady gas flow up through a borehole section with no penetration below $d_e$, two transport mechanisms
can be considered, convection or molecular diffusion. As for convection, no driving force to induce a flow in a water
column in a borehole, in particular no significant temperature variations, has been observed in the Hartoušov borehole.
The mass flux due to molecular diffusion, on the other hand, can be estimated as follows, and it appears to be very
limited. Assuming that the concentration of the dissolved $CO_2$ in the resting water column increases with increasing
depth as much as allowed by the increasing hydrostatic pressure (with the Henry's law constant being of the order of
$10^{-5}$ kg m$^{-3}$ Pa$^{-1}$, see Sander, 2015), then the corresponding diffusive flux (with the diffusivity being of the order of $10^{-9}$ m$^2$s$^{-1}$) through the borehole of the given cross-section area (say, $10^{-2}$ m$^2$) would be no more than of the order $10^{-12}$ kg

s$^{-1}$. We thus infer that the gas bubbles enter the Hartoušov borehole in its penetrated section, as we assumed in the
previous section.

**2.5 Laboratory test of bubble fraction method**
The methodology for indirect gas flow measurement using pressure difference was first tested in the laboratory. The
experimental setup consisted of an air pump connected with a valve for controlling the air flow, which was led to the
bottom of a plastic tube with an inner diameter of 10.5 cm and a height of 2.5 m simulating the borehole. Two water
pressure probes were installed at a fixed distance of 0.5 m on a vertical rod inside the tube and all the air from this
tube was led to the chamber gas flowmeter. The air inflow was increased stepwise, and at each level the data were
recorded for a period of 15 min using a 1Hz sampling rate. The gas flow ranged from 0 to 30 L/min, which corresponds
to the volumetric flux ranging from 0 to 0.06 m/s. The observed mean bubble fraction appears to increase nonlinearly
with the gas flow. Note that the modification using (Eq. 4) is insignificant here, due to the fact that both pressure
probes are at a depth of less than 1 m. The bubble fraction values are more scattered than the gas flow rate measured
by the flowmeter. The resulting noise was partially suppressed by low pass filtering of the pressure data using an RC
circuit with a time constant of 30 s and additionally by 1 min data sampling to smooth the water level values.
It is worth noting that the dynamics of bubbly flow in a borehole is quite a complex issue, which however appears to
have been studied rather intensively in the chemical and nuclear engineering literature (see, e.g., Ghiaasiaan, S.M.,
2008; Montoya, G. et al., 2016). The simple considerations introduced in the previous text (Eq. 3 and 4) correspond
to the drift-flux model for a vertical borehole, provided that the water flux through the borehole is negligible. In
particular, any momentum exchange with the walls is ignored. While this approach is well justified for the bubbly
flow regime observed with small gas fluxes, with increasing volumetric gas one observes different flow regimes of
greater complexity, such as the slug flow and churn flow. As the bubbles ascend, they increase in volume, join each
other merrily or even sadly split apart, their brief life being eventually cut short by obstacles such as the pressure
probes dangling in the well; these are however out of the scope of this paper. Even in the bubbly flow regime, the
relation between the bubble fraction $\phi$ and the volumetric flux of the gas bubbles $u$ [m/s] has been described, e.g., by
the following well established empirical relation (Zuber, Findley, 1965).





$$\phi = \frac{u}{C_0 u + V_{gj}} \qquad (5)$$
where $C_0 = 1.13$ and, assuming that the density of the gas is negligible when compared to that of water,
$$V_{gj} = 1.41 \left(\frac{\sigma g}{\rho_w}\right)^{1/4}$$
The curve in Fig. 5 is obtained by taking the surface tension for water and air at the laboratory temperature as
$\sigma = 0.07$ N/m. It appears that the mean bubble fractions derived from our pressure probe data overestimate the void
fractions given by the Zuber-Findley model, in particular for low flow rates. For comparison, we append to Fig. 6 the
projected bubble fraction data $\phi_0$ plotted against the corresponding gas flow rates measured in the Hartoušov site. The
comparison to the laboratory data and to the Zuber-Findley model reveals a discrepancy that indicates some need for
further analysis, which is beyond the scope of the present study. Let us briefly note that the difference cannot be
explained by the mere parametric differences from the laboratory setting such as the temperature, gas and water
composition.
While it appears that the projected bubble fraction data from the Hartoušov site cannot be directly inverted to obtain
a reasonable gas flow rate estimate, it is important that they provide a fair correlation (see also Fischer et al. 2017 and
Fig. 4 in their paper) and can thus provide a valuable gas flow rate proxy. As expected, the integral method data seems
to perform better than the less recent differential data (see Section 2.3). Note particularly the high noise of the latter
and its lower correlation to the gas flow rate measurement (Fig. 6). Accordingly, using the pressure sensors at a larger
distance, and, if possible, placing one of them below the bubble entry depth, seem preferable for indirect gas flow
measurement.
*Figure 5*
*Figure 6*

**2.6 Environmental effects**
The measurements of $CO_2$ flow, $CO_2$ pressure and pressure head are influenced by environmental effects – mainly
variations of temperature (diurnal, seasonal), changes of barometric pressure and tidal effects. Temperature and
barometric changes are the most significant, since their influence can be local and can vary even among the stations
of the network. Barometric and tidal loading of aquifers has been studied in detail (e.g. Jacob, 1939; Rojstaczer and
Riley, 1990; Roeloffs, 1996). Here, we address the basic principles that are relevant to the pressure and production of
the upstreaming gas. Both the confined and unconfined response of pressure head are characterized by the barometric
efficiency $E_B$, which expresses the ratio of the change of the hydraulic head $\Delta h$ caused by the barometric pressure
change $\Delta b$
$$E_B = \rho_W \, g \frac{\Delta h}{\Delta b} \qquad (6)$$
The net response is always a decrease in the hydraulic head with an increase in barometric pressure. The barometric
pressure variations act directly on the open water level in the well and also on the formation composed of the mineral
matrix and pore space filled by the water. As a result, the direct effect on the water level in the borehole is partially





suppressed by the fraction of the external load borne by the formation water. Hence, the barometric efficiency can
also be written as
$$E_B = \frac{\theta\beta}{\theta\beta+\alpha} \tag{7}$$

where $\alpha$ and $\beta$ is the compressibility of the rock matrix and water, respectively, and $\theta$ represents porosity within the
aquifer. Thus, barometric efficiency can be described as the fraction of specific storage derived from the
compressibility of water or, equally, as the fraction of external load change borne by the formation either as
compaction or expansion. Accounting for the range of fractured rock compressibilities, $E_B$ of confined aquifers usually
ranges between 0.2 and 0.7 (Todd, 1980) and may reach 1.0 for granite with a very low compressibility of the rock
matrix (Roeloffs, 1996; Acworth and Brain, 2008). Note, however, that large values of $E_B$ may as well correspond to
large values of β; this fact is not addressed in the literature for the simple reason that it is usually the rock that varies
from site to site and not the water. In this concern, the possible effect of the presence of the compressible $CO_2$ bubbles
within the aquifer surrounding the borehole on the barometric efficiency is a question that has not been addressed in
the literature, to the best of our knowledge.
Similarly, the barometric effect to the $CO_2$ discharge from an aquifer through an open well has not been studied either.
One can expect that an increase in the pore pressure due to an increase in the barometric pressure allows for larger
amounts of $CO_2$ to be dissolved in water, which in turn decreases the volume of $CO_2$ leaking into the well. Similarly,
a decrease in barometric pressure may induce increased degassing. Note that there exist many unknowns in this regard,
such as the flow paths of the gas ascending through the aquifer, the amount of the mobile and immobile gas bubbles
in the porous space, etc. In the Hartoušov borehole, a strong anticorrelation between the gas flow rate and the
barometric pressure has been observed.
Other external effects like diurnal temperature variations and Earth tides were found much weaker than the influence
of barometric pressure. The volumetric fraction of bubbles is not affected by air temperature, since the sensors are
placed in groundwater with an almost constant temperature. In addition, the periods of diurnal temperature variations
and significant Earth tide components are significantly shorter than the expected durations of anomalies of deep-
generated gas flow. Accordingly, we do not apply corrections for temperature variations and Earth tides.
We correct the measured quantity $f$ (pressure head or gas flow) for demeaned barometric pressure variations $b$ using
the equation $f_c = f - E_B (b - <b>)$. Barometric efficiency $E_B$ is determined with the target of minimum cross correlation
of $f$ and $f_c$. To account for the possible time variation of $E_B$ a sliding window of one day is used; see Fig. 7a for original
and corrected records of pressure head and gas flow in Hartoušov. Fig. 7b shows the cross-correlation functions
between barometric pressure and original and corrected records. The success of barometric correction is indicated
both by removing the anti-correlation with air pressure and by minimizing short period variations in the corrected
records. The mean barometric efficiency was 0.76 for the pressure head and 0.46 L/min/kPa for the gas flow.
*Figure 7*





**3. Results of CO₂ flow monitoring in West Bohemia**

The time series of gas production at all monitored stations, along with seismicity plot, are shown in Fig. 8. The record at Hartoušov for the period from late 2007 to 2019 shows a long-term decrease interrupted by several abrupt massive increases in gas production. The maximum flow, reaching 50 L/min, followed the 2014 seismic sequence in late summer/autumn 2014; the minimum values, below 10 L/min, were observed prior to the 2014 seismic sequence and at the present time. The fast coseismic increase and long-term postseismic decrease are visible both in the gas flow and *integral bubble fraction* data determined using eq. (3) and (4) and are consistent with Sibson's fault valve model (Fischer et al., 2017). Note particularly the abrupt rise in gas flow and CO₂ bubble fraction during the $M_L$ 4.4 seismic sequence of May – August 2014 and in bubble fraction during the October 2008 $M_L$ 3.8 swarm. Next to these striking coincidences of seismic activity and CO₂ release we also find cases of strong seismic activity, which was not accompanied by a significant gas flow anomaly (see the $M_L$ 3.4 swarm of 2011 and the most recent $M_L$ 3.8 swarm of 2018). On the other hand, the CO₂ flow record shows a few positive pulses which are not related to significant seismic activity (Fig. 8b). The most striking one is the gas production increase in the period from the beginning of May till the end of July 2016, which is visible both in the gas flow and bubble fraction records. This is, however, undoubtedly of anthropogenic origin caused by drilling of the nearby 108 m deep HJB-1 borehole at a distance of 40 m from the monitored VP 8303 borehole, which drilling started on March 30 (Bussert et al., 2017). The drilling reached the ceiling of a CO₂ pressured horizon at a depth of 80 m on April 21 and created a shortcut to the shallower aquifer, which was tapped by the monitored borehole. The three-month long gas increase thus represents a delayed response to a nearby drill. Another, less pronounced, positive pulse in the period from mid-September to late November 2016 is of unknown origin. A number of negative pulses and oscillations are found on the bubble fraction record alone, which lower the correspondence between the gas flow rate and the bubble fraction data and indicate a more complex relation between gas flow in a borehole and volume fraction of ascending bubbles, as already noted in section 2.5.

The records of gas *differential bubble fraction* data in Bublák and *resistivity-based bubble fraction* in Soos indicate in the monitored period since autumn 2015 a steady gas release with only a few bumps, which are most probably of local origin and related to the shallow character of the mofettes. Gas at these sites passes through approximately cylindrical vents of ~0.5 m diameter and ~1 m depth filled by surface water. The similarity of bubble fraction increase at Soos and gas flow increase at Hartoušov in summer 2016 is most probably merely accidental, considering the anthropogenic origin of the rise at Hartoušov and the large mutual distance of about 5 km of these sites.

As mentioned in section 2.3, the integral method of bubble fraction measurement provides better results than the differential method. The latter suffers particularly from high noise caused by the placement of both pressure probes in a water column with flowing bubbles as shown in Fig. 5. One can also notice a better correlation of the integral method compared to the differential one. Unfortunately, due to technical problems, we were not able to perform this comparison for the same time window – so time windows of the same length (3 months) free of any technical issues were selected.



## 4. Discussion

The *barometric efficiency $E_B$* of the groundwater pressure head of 0.76, which we obtained, is relatively high. The high values of $E_B$ are generally considered an indication of the small compressibility of the rock matrix that is typical for unweathered granite (Acworth and Brain, 2008). The target aquifer is formed by sedimentary formations of the Cheb Basin composed of sandstones and conglomerates with varying clay contents underlain by mica-schist basement (Bussert et al., 2017). The compressibility of these types of rocks is, however, 3 to 6 times greater than of granite (their bulk moduli range from 10 to 20 GPa compared to 50 GPa for granite). Using Eq. (4), porosity 30% and bulk moduli ratio of matrix and pore fluid equal to 5, one gets $E_B = 0.5$. The level $E_B = 0.76$ is reached for bulk moduli ratio of 15. Assuming the bulk modulus of aquifer rocks about 10 GPa, one obtains a bulk modulus of the fluid of only about 0.7 GPa, which corresponds to three times larger compressibility than for water. This could be explained by the presence of carbon dioxide in the groundwaters in gaseous phase and is worth further research.

*The gas flow trend* in Hartoušov after the 2014 seismic sequence shows signatures similar to those in the period before 2014, which followed the 2008 swarm. A similar, long-term overall decrease is followed by steady state behavior with an almost constant flow rate of about 10 L/min. In terms of the Sibson's fault valve model, this corresponds to the self-sealing phase of the fault due to mineral precipitation (Sibson, 1992) when pressure builds up and in combined action with tectonic loading results in increasing instability of the fault. This inevitably leads to later recurrence of fault failure in the form of seismic activity and regeneration of fault permeability. As indicated above, the coincidence of a massive rise in $CO_2$ flow and seismic activity has not been observed since the 2014 seismic sequence. Indeed, none of the earthquake swarms since 2014 have been accompanied by a distinct $CO_2$ degassing anomaly (Fig 8b). All in all, in the whole period of $CO_2$ flow monitoring in Hartoušov since 2007, five earthquake swarms with magnitude $M_L$ larger than 3.0 occurred (2008, 2011, 2014, 2017, 2018) and only two of them (2008 and 2014) were accompanied by a strong and long-lasting coseismic increase in $CO_2$ degassing. This is not surprising in general, because the fault valve mechanism might act only under certain circumstances. And even if a fluid pulse is released during every stronger seismic sequence its volume might not be sufficient to reach the Earth's surface with a detectable amplitude. This is also directly related to the pressure buildup in the fluid reservoir beneath the sealed fault, which is a long-lasting process and thus earthquakes that occur soon after releasing the accumulated fluid pressure are likely to not be accompanied by a significant fluid release. Recently, since the summer of 2019, the $CO_2$ flow rate in Hartoušov has decreased below 10 L/min, which could be a sign of the approaching occurrence of a new seismic swarm, according to the Sibson's fault valve model.

In this context it is also of interest to consider the *hypocenter cluster geometry* in 3D and its relation to the presence of permeable channels in a shallow crust allowing crustal fluids to reach the surface. In Fig. 9 hypocenters of individual earthquake sequences are indicated in a vertically oriented cross section and show a hat-like structure in depths from 6.0 to 10.5 km extending about 10 km north-south. The Hartoušov mofette field is located about 10 km south from the center of the main cluster, which corresponds to 6 km distance from its southern tip. A pronounced segmentation of the fault plane is apparent with the 2008 and 2014 segments in the southern branch of the cluster and the 2011, 2017 and 2018 segments clusters in its northern branch. It is worth noting that the 2014 mainshocks showed unfavorable oriented focal mechanisms and occurred on a fault jog activated by stress concentration resulting from



previous swarm activity (Hainzl et al., 2016, Jakoubková et al. 2018). This structural and possibly impermeable
boundary within the fault zone was broken by the $M_L$ 3.5 mainshock of the 2014 sequence - the first earthquake of
this sequence which was followed by the massive $CO_2$ flow rise in Hartoušov.
The clear coseismic $CO_2$ flow rate increase during the 2008 and 2014 seismic sequences indicates the presence of a
*permeable channel between the southern cluster and the Hartoušov mofette field* (Fig. 9). The absence of $CO_2$ flow
anomalies coinciding with the seismic activity in northern clusters could be interpreted to show that the hydraulic
connection between these fault patches and the Hartoušov mofette is missing, which could be related to the afore-
mentioned fault jog. Besides, it is also of interest that epicenter distribution and $CO_2$ degassing occurrence is typically
separated in the area (Weinlich et al., 2006; Babuška and Plomerová, 2008); most earthquakes occur in the northern,
$CO_2$-free part of the Cheb Basin.
*Other monitored sites* such as Bublák and Soos show, similar to Hartoušov, almost constant $CO_2$ discharge since early
2017. As these stations were not in operation during the 2008 and 2014 seismic sequences showing coseismic $CO_2$
increase in Hartoušov, no inferences about their correlation to the seismic activity can be drawn. Continuous
monitoring of $CO_2$ degassing is required to determine whether future seismic activity in the southern cluster will
generate an increase in degassing in either of the monitored sites and enable the verification of the hypothesis that
only earthquakes in the southern cluster are capable of generating a $CO_2$ pulse which reaches the surface.
**5. Conclusions**
The present study is focused on the long-term monitoring of $CO_2$ degassing in the form of mofettes and gaseous
mineral springs targeted on the West Bohemia/Vogtland region in Central Europe, which is typified by the occurrence
of earthquake swarms and emanations of carbon dioxide of magmatic origin. The gas flow measurement is applied to
two types of sources: natural wet mofettes with gas outflow through surface water pools and boreholes tapping shallow
$CO_2$-saturated aquifers. The different local conditions of the five monitored sites call for different methods of gas
capture and flow rate measurement. Besides the direct flow measurement using a drum chamber gas flowmeter,
electronic MEMS flowmeters and Venturi-based probes we introduce a novel, indirect method based on quantifying
the gas bubble contents in a water column, which is capable of functioning in severe environmental conditions. The
method is based on measuring the pressure difference along a fixed depth interval in a water column, which is
proportional to the mean bubble fraction within the measured section. We analyze the dependence of the bubble
fraction on depth and project it to the atmospheric pressure to make it directly comparable to the gas flow rate.
Laboratory tests indicate the nonlinear dependence of the bubble fraction on the flow rate, which is confirmed by
empirical models found in the chemical and nuclear engineering literature. Flow rates and bubble fractions observed
in a pilot borehole in the Hartoušov mofette show a high mutual correlation, however some discrepancy is found
between the measured flow rate and that predicted by the empirical models. This discrepancy calls for further
analysis.
We also analyzed the long-term monitoring of gas flow and bubble fraction in the pilot borehole for the period 2008
– 2019. We found a quite strong barometric influence on the hydraulic head of the confined aquifer corresponding to



a barometric efficiency of 0.76, which can be attributed to the compressibility of the pore fluids including the gaseous
phase of carbon dioxide.
The record of gas flow rate and bubble fraction in Hartoušov shows two high-amplitude coseismic rises coinciding
with the occurrence of earthquake swarms in 2008 and 2014. The flow rate increased to a multitude of the preseismic
level for several months and was followed by a long-term decay. However, another three seismic swarms occurring
in the same fault zone were not associated with any significant $CO_2$ flow anomaly. We surmise that this may be related
to the slightly farther location of hypocenters of these swarms in comparison with the two which caused the coseismic
$CO_2$ flow rise. Further long-term CO2-flow monitoring is required to verify the mutual influence of $CO_2$ degassing
and seismic activity in the area.
**Code/Data availability**
Most of the data analyzed in the manuscript including email address for requesting additional data are available
online at web.natur.cuni.cz/uhigug/carbonnet/en_index.html .

**Author's contribution**
VB and TF designed and carried out the measurements and ML formulated the theoretical part with support of TF. TF
prepared the manuscript with contributions from all co-authors.

**Competing interests**
The authors declare that they have no conflict of interest.

**Acknowledgments**
CO2 flow monitoring and the work of the authors was supported by the project CzechGeo/EPOS-Sci
(CZ.02.1.01/0.0/0.0/16_013/0001800). The authors thank also to Jan Vilhelm for valuable ideas.



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



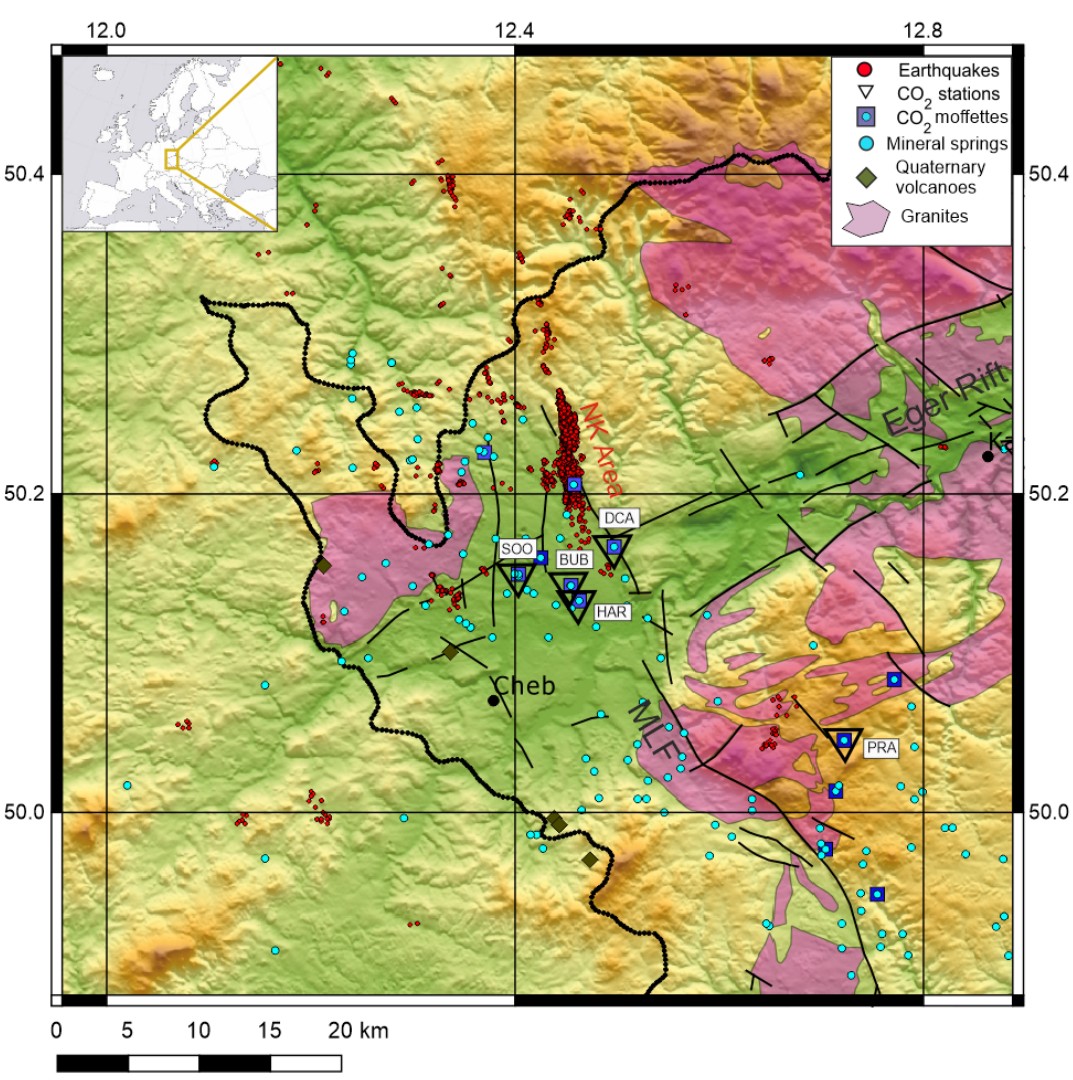


**Figure 1. Relief map of the West Bohemia/Vogtland region with the fault network, granite units, and major fault zones as Mariánské Lázně Fault (MLF) and Eger Rift. Seismic events, major focal zone of Nový Kostel (NK Area), CO₂ monitoring stations, CO₂ mofettes and mineral springs and Quaternary volcanoes are also indicated (see the legend).**







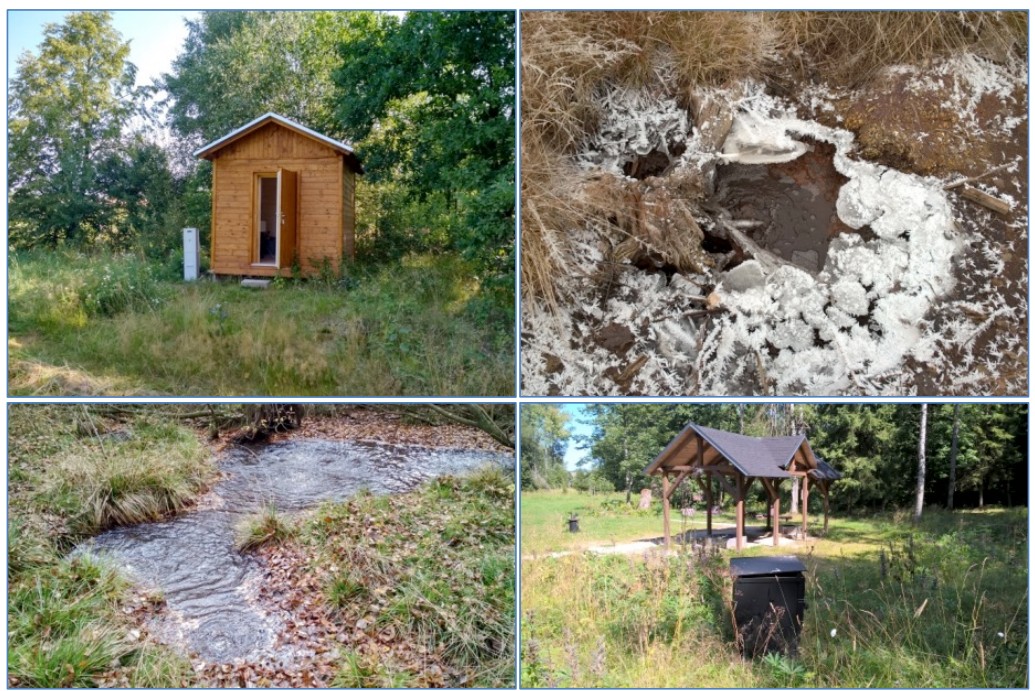


**Figure 2. Photos of selected CO2 monitoring fields – Hartoušov, SOOS, Bublák and Prameny**








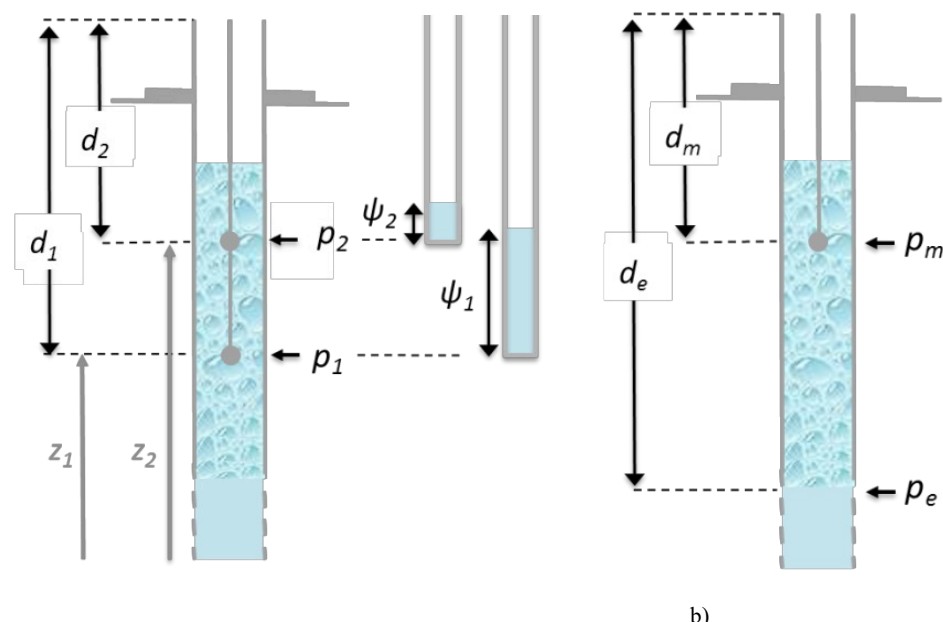


a)                                                                                      b)

**Figure 3. Measurement of pressure in the borehole with ascending bubbles: the measured pressure p and the related pressure head ψ at two different depths $d_1$ and $d_2$ within the bubble column using the *differential method* (a); pressure within the bubble column $p_m$ and beneath $p_e$ used to determine the mean bubble fraction using the *integral method* (b). Note that the difference of altitudes $z_2 - z_1 = d_e - d_m$.**







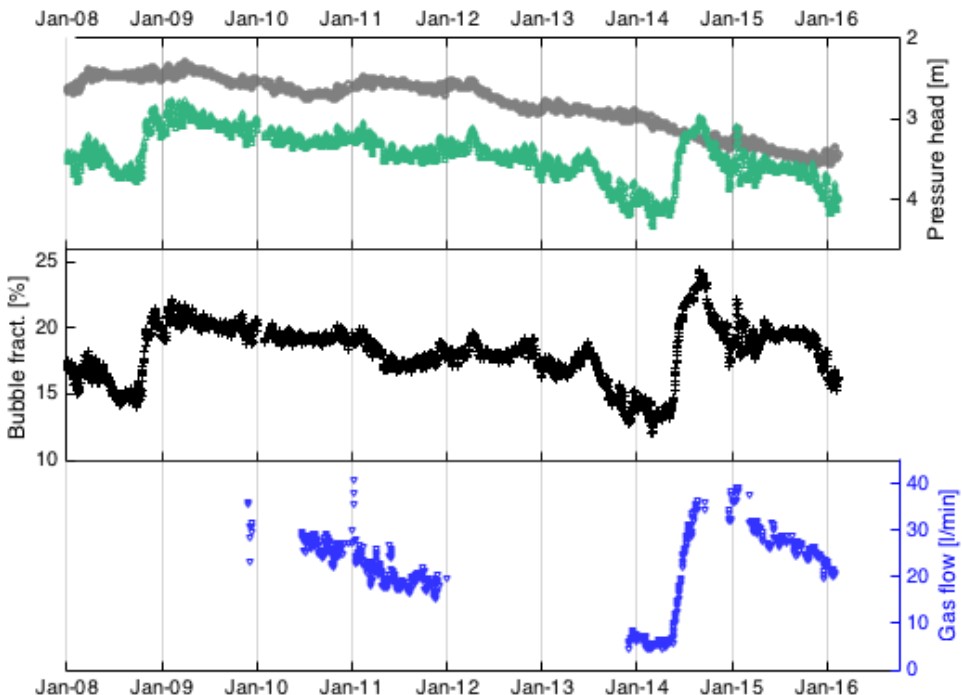


**Figure 4. Daily averages of pressure head $h_e$ in the reference well (grey) unaffected by gas flow and in the monitored Hartoušov well, $h_m$ (green); the volumetric fraction of bubbles (black) determined by Eq. 4 and $CO_2$ flow (blue) in the Hartoušov well for the period 2008-2016.**






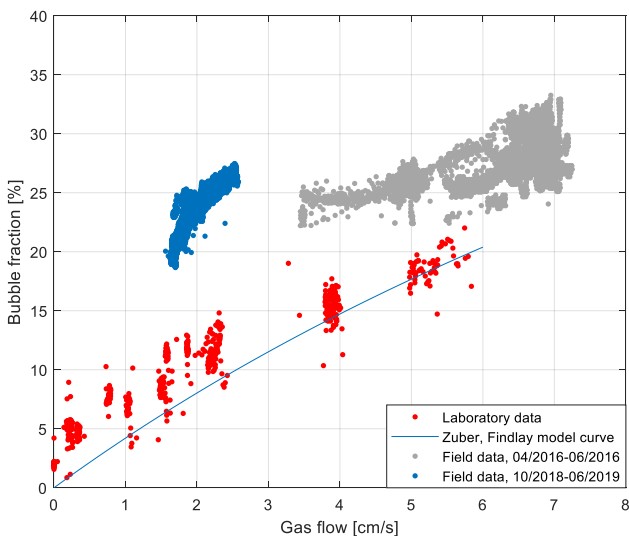

**Figure 5. Comparison of gas flow and volumetric fraction of bubbles for the field (Hartoušov mofette) and laboratory measurement. Laboratory measurements are smoothed by RC circuit of 30 s time constant (red points) and additionally by running average of 1 min length; the field measurements from 2016 (grey points) are based on the differential method with sensor distance of 1.0 m; the field measurements from 2018-2019 (blue points) are based on the integral method. Blue line is the best fit based on eq. (5).**

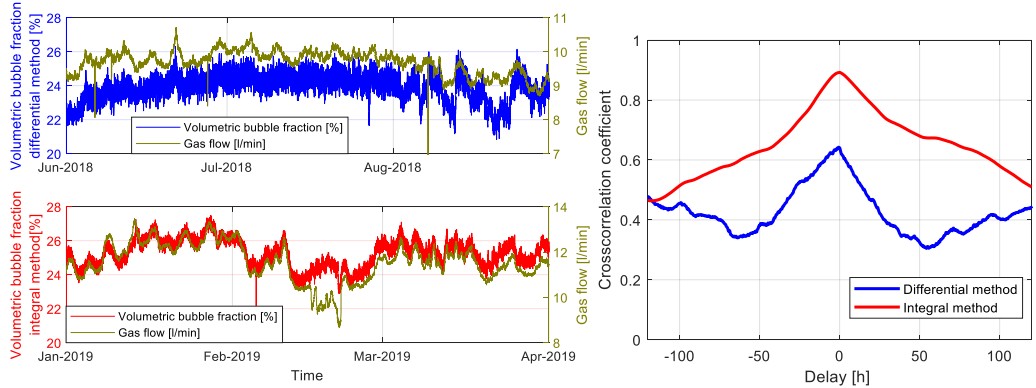

**Figure 6. Comparison of volumetric bubble fraction derived using integral and differential methods with the gas flow at the Hartoušov site. Depth of pressure probes for the differential method are 4.45 and 5.45 m below the surface. For the integral method it is 5.45 m and 27.2 m.**






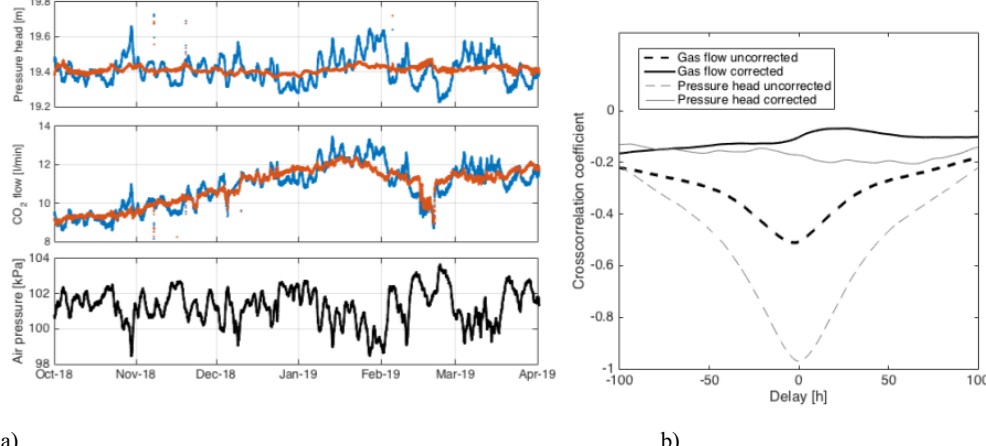


a)                                                                                      b)

**Figure 7. The barometric effect to the pressure head and CO$_2$ flow in the Hartoušov borehole for the period from October**
**2018 to April 2019. a) Original measurements are indicated in blue and those corrected for barometric pressure are in red.**
**The upper panel shows pressure head at the depth below the bubble formation and the lower panel shows gas flow measured**
**by flowmeter. The success of barometric correction is illustrated in b) showing the decrease of barometric anticorrelation**
**after correcting.**





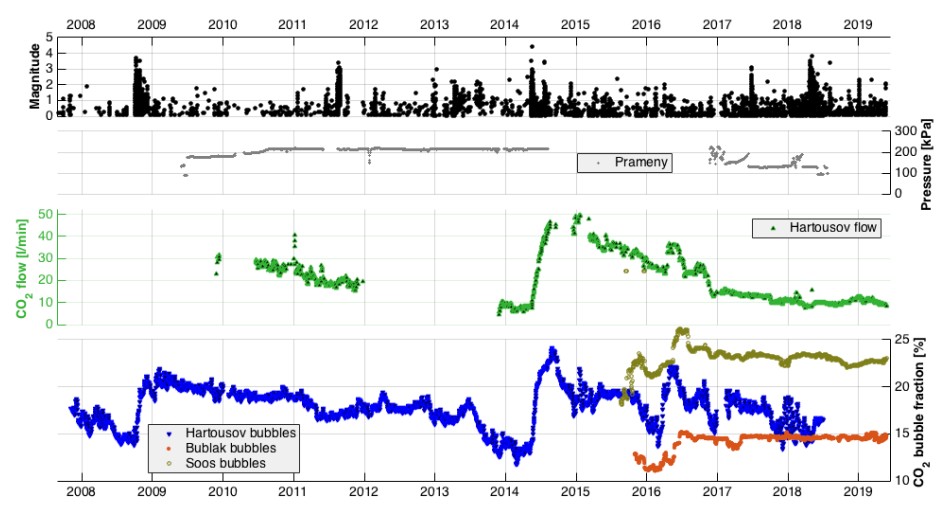

a)

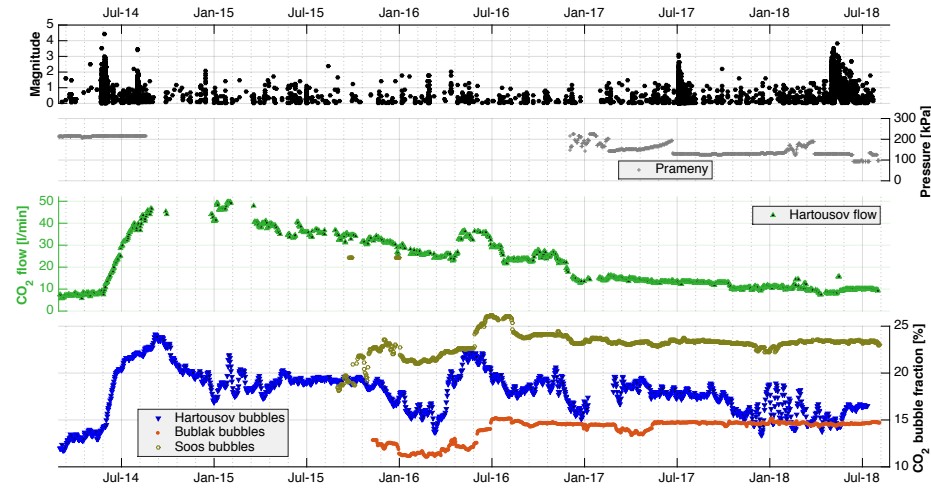

b)

**Figure 8. Comparison of seismic activity and CO₂ production at individual monitoring sites in West Bohemia. For Hartoušov the CO₂ flow and gas bubble fraction are shown; for Bublák and Soos the CO₂ bubble fraction is shown and for Prameny the gas pressure in a closed borehole is plotted; (a) period 2007-2019; (b) detail for the period 2014-2019. Gas bubble fraction was determined using the integral method.**

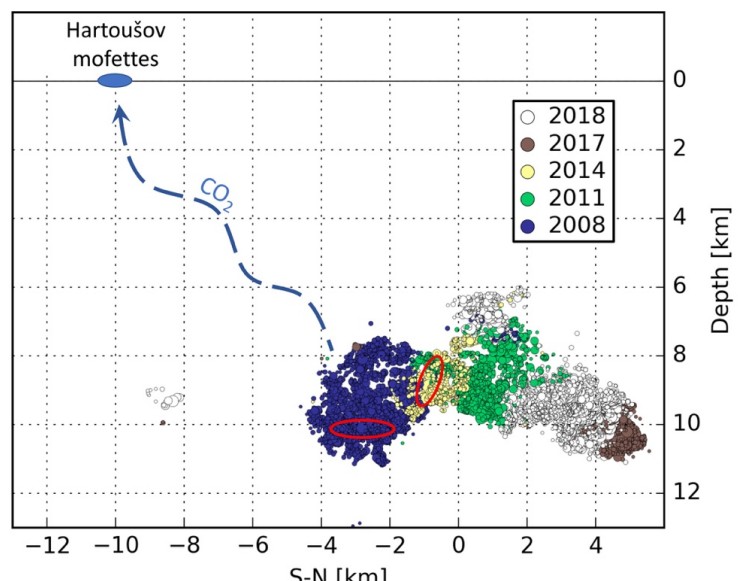

708

**Figure 9. Vertical, north-south oriented section through the Nový Kostel fault zone with hypocenters of the earthquake swarms occurred between 2008 and 2018. The position of the Hartoušov mofette showing coseismic CO$_2$ flow rate increase is indicated on the surface. Red ovals show the position of first events of the 2008 and 2014 seismic sequences.**