# Peer review of "Monitoring crustal CO2 flow: methods and their applications to"

_Solid Earth, 2020_

## Referee Comment (RC1) · Anonymous Referee #1 · 15 Mar 2020

Monitoring crustal CO2 flow: methods and their applications to the mofettes in West Bohemia By Tomáš Fischer, Josef Vlček, and Martin Lanzendörfer

General comments The manuscript is of principal interest for the scientific community, not only for the West-Bohemia area with their numerous CO2 gas emission sites, but generally for other geothermal/volcanic areas because it proposes a possible observation technique of natural degassing sites like mofettes or CO2 enriched mineral springs. I have no specific suggestions concerning the scientific content of the manuscript; however, some points should be improved before publication.

The authors document a comprehensive analysis and discussion of robust monitoring methods of CO2 degassing sites which were compared during long-term monitoring periods. They invoke a simple but applicable installation of pressure sensors to reg-

ister the gas flow rate via a relative pressure ratio. The influence of the CO2 gas bubbling process to the groundwater column in wells or mofettes was registered and can be converted into a gas flow rate as a reliable measuring technique. Influences of barometric pressure and ambient temperature are discussed. More experiences in the data evaluation and some improvements are necessary as mentioned at the end of the studies but the application on other areas of interest can be suggested. One of the most interesting aspects of this study is the relation to the earthquake swarm occurrence north of the gas emission sites and their triggering of gas flow anomalies. The results suggest a trigger process for gas flow anomalies only for earthquake swarms in a southern cluster. Swarms north of it show no influences to the gas emission. This result implies a seismotectonic influence to the gas migration which is important for the interpretation of gas flow anomalies discussed worldwide. This is an interesting paper that should be accepted for publication following consideration of the reviewers' comments which are marked here with the line numbers but they are all quite minor.

Specific comments 38: Fig. 1 : The map: the marked distribution of granite is not complete. And it is a questionable presentation: what is important: relief or lithology- I suggest only one of them. The northern most mofettes in Fig. 1(latitude > 50.2)? They are really mofettes? Please check it.

56-62: at the end of these sentences, the authors should add here the sentences of the lines 73-81 for a better overview about the gas isotopic features. 61: what is the meaning of "... high 3/4 He ratios..." ? 65: similar as above: What is high gas flow ? , see also line 73 and 75: unclear. 67: "the ascent of gas" – Numerous studies show that the earthquake swarms are related to the ascent of gas? This assumption has no evidences in my opinion or please, indicates the references. 73 "gases produced" , this terminus is may be correct but not usual for the characterization of natural degassing sites, see also line 108 & 405 104: please add: (see Fig. 1) 110: what means "deep root" zone of mofettes? The origin of CO2 is known. 201 "Within a few months" should be changed into " Within a few days..." ? 294 & 295: This explanation is may be correct

for mineral springs with a continuous gas/water discharge. However, mofettes can be considered as gas dominated migration path. It means that the $CO_2$ will migrate as gas phase with over-pressure above the supercritical point. The water phase content will be of minor importance here. The groundwater horizons are the barriers and the beginning of the bubble creation which depends on the pressure ratios of gas and the water column if the maximal solubility in water is reached. This effect can be observed in submarine gas vents. 300: the driving force for gas flow is the hydraulic pressure gradient and the density contrast 354: the section 2.6: the interpretation with the barometric efficiency is an interesting approach. Because of "the many unknowns in this regards", a simplified way could be helpful in this context. What about this comparison: show an additional graph with the result of pressure head (in mbar) minus the atmospheric pressure (in mbar). 387-391 these lines should be at the end of this section. 427&428: The authors claim the increase of gas discharge as anomalous effect of different reasons except the anomalies as "probably merely accidental" at two sites (Soos and Bublák) during the summer 2016. These anomalies occurred a few weeks after the gas flow increase at Hartoušov due to the drilling process and the influence to the hydraulic regime. An assumption or specific interpretation should be added. Please think about the fluid interaction of the deeper horizons in the area (Cheb basin). For example, the gas eruption at the drill site H11 in the year 1957 induce an anomalous gas discharge and variations in the water levels in Františkovy Lázně, about 2 km far. A reduced water table at the Hartoušov drill site could influence the hydraulic pressure regime in the nearby Cheb basin. This influence could trigger the ex-solution of $CO_2$ of the water table with a temporal delay at other locations (mofettes), similar to the atmospheric pressure effect. 464: Please take into account also that the strong drought period during the last summer reduces the level of the surrounding ground water table. This hydraulic pressure reduction induces an addition gas release as diffuse component and could reduce the total amount of gas discharge at the monitoring site. 477: because of missing evidences of this process, please add "... indicates the possible presence of..."

Technical comments CO2 – should be written with 2 subscript The names of the references in the text should be outside the brackets, e.g. Fischer et al. (...), see line 68, 250, 259 a.s.o. 439: considered as an 493 "discharges" is a better term here than "emanations" 564 this reference is not mentioned in the text A few typesetting mistakes in the reference list The figure captions should be not in bold

All the 15 listed questions for the reviewer can be answered with: accepted or yes 1. Does the paper address relevant scientific questions within the scope of SE? 2. ...

---

## Referee Comment (RC2) · Anonymous Referee #2 · 6 Apr 2020

The paper discusses different methods to monitor crustal $CO_2$ flow, in this case specifically for the West-Bohemia area. The content is of interest to the scientific community and based on a comprehensive collection of long-term monitoring data collected with various methods at different test sites. A main focus is given to the estimation of gas bubble fractions from pressure measurements. What is missing in my opinion is a short summary statement at the end of the method section, how these measurements compare to the other methods utilized in the area. For example a statement about the performance of resistivity measurements, which were also used to estimate the gas bubble fraction although in a different set-up, would be interesting. In general, the paper is well-written and structured, however some clarifications would be beneficial. My suggestions are all minor and listed below by page and line number:

[Figure]

Interactive
comment

P2 L50: Could you please specify which fault zone is meant here?

Table 1. CarbonNet monitoring network: Hartoušov, the 105.8 m deep borehole is not mentioned in the description of the network in the main text (unless I missed it). Is the borehole used as the reference borehole for the integral method described in the following?

P6 L183: "ceiling of the aquifer" the term seems strange to me, I assume it means simply the top of the aquifer.

P9 LL 260 – 264: ".... that the gas bubbles have to appear at the penetrated section of the Hartoušov borehole. This allows us to determine the mean volumetric fraction of the bubbles using eq. (3) with $h1(t) = hm(t)$ being the hydraulic head measured at the depth $dm = 4m$, and $h2(t) = he(t)$ being the hydraulic head measured in any depth below the bubble entry depth, which we suppose to be at the upper part of the penetrated section at $de = 20.5m$ (Fig.3)." The statement is confusing to me. If the gas bubbles enter the borehole at the penetrated section, how can the upper part of the penetrated section be below the bubble entry depth? Is the hydraulic head for $he$ actually measured in the Hartoušov borehole or in the reference well mentioned before? I assume the latter is the case according to Figure 4. Could you please clarify the text here?

P10 L 307: What is the observed mass flux at the teste site?

P 10: Section: Laboratory test of bubble fraction method: Could you please state
clearly here which bubble fraction method was tested in the laboratory, the integral or the differential method? I assume the latter is the case.

P 11 LL 346 – 350 (Section: Laboratory test of bubble fraction method): I assume the statement, that the integral method performs better than the differential method, is purely based on the observed correlation of the field data and not supported by the laboratory tests. Although, I agree with the statement it seems to be a bit out of context here. Furthermore, why is a different time window utilized for the differential method in Figure 5 and Figure 6? Maybe a separate section discussing the differences in more detail would be better here including the statement on P13 LL 432 – 436, which I assume refers back to Figure 6 and not Figure 5.

P12 LL 385-386: This statement should be followed by paragraph P12 LL 392-399. The small rearrangement would make it easier for the reader to follow, that there is a large effect on the data due to barometric pressure variations and that these have to be corrected. Maybe that could also be explicitly mentioned, although it is implicitly clear.

P8 L283 and 243: $\phi_o$ should be capitalized
P10 LL 319-321: Figure 5 should be referenced here.
P11 L338: This should be Figure 5 and not Figure 6.

---

## Author Comment (AC1) · 16 Apr 2020

Thank you for your helpful comments. We address all of them in our replies below and in most cases have modified the manuscript accordingly, which is in each case indicated in our reply.

Ad Specific comments

38: Fig. 1: The map: the marked distribution of granite is not complete. And it is a questionable presentation: what is important: relief or lithology- I suggest only one of them. The northern most mofettes in Fig. 1(latitude > 50.2)? They are really mofettes? Please check it.
*important: relief or lithology- I suggest only one of them.*

56-62: at the end of these sentences, the authors should add here the sentences of the lines 73-81 for a better overview about the gas isotopic features.
*Thank you for this suggestion, we moved the paragraphs accordingly, which also allowed to remove one sentence.*

61: what is the meaning of "... high 3/4 He ratios..." ?
*Now, after rearranging the paragraph, it gets clearer.*

65: similar as above: What is high gas flow ? , see also line 73 and 75: unclear.
*We have indicated the daily discharge of dozens of tons of gas*

67: "the ascent of gas" – Numerous studies show that the earthquake swarms are related to the ascent of gas? This assumption has no evidences in my opinion or please, indicate the references.
*Our formulation reads precisely "numerous studies of the local earthquake swarms show that they may be related to pressurized fluids in the crust and the ascent of gas". First, we mention fluids in general and only then gas. And we cite these studies in the next sentence. So, we decide to keep the sentence as it is.*

73 "gases produced", this terminus is may be correct but not usual for the characterization of natural degassing sites, see also line 108 & 405
*Thank you, we changed to discharge (the first occurrence exists no more because the sentence was removed)*

104: please add: (see Fig. 1)
*OK, added.*

110: what means "deep root" zone of mofettes? The origin of CO2 is known.
*Yes, the gas is most probably of mantle origin. But here, we have in mind the mofettes, compared to diffuse degassing. In particular the fast coseismic reaction of the mofettes, which indicates that the CO2 origin is deeper than hypocenters, but nothing about the mantle origin.*

201 "Within a few months" should be changed into " Within a few days..." ?
*No, the onset of increase started 4 days after the first event (mainshock) and continued for about three months, which underlines its significance.*

294 & 295: This explanation is may be correct for mineral springs with a continuous gas/water discharge. However, mofettes can be considered as gas dominated migration path. It means that the CO2 will migrate as gas phase with over-pressure above the supercritical point. The water phase content will be of minor importance here. The groundwater horizons are the barriers and the beginning of the bubble creation which depends on the pressure ratios of gas and the water

column if the maximal solubility in water is reached. This effect can be observed in submarine gas vents.

*Thank you for this comment; we added a sentence "or in the presence of significant water discharge, such as in mineral springs." to the end of the first paragraph of 2.4 to take this into account.*

300: the driving force for gas flow is the hydraulic pressure gradient and the density contrast
*Thank you for the comment; we had in mind "steady flow of the dissolved $CO_2$" and have modified the text accordingly.*

354: the section 2.6: the interpretation with the barometric efficiency is an interesting approach. Because of "the many unknowns in this regard", a simplified way could be helpful in this context. What about this comparison: show an additional graph with the result of pressure head (in mbar) minus the atmospheric pressure (in mbar).
*The reviewer is right that such a plot might be illustrative. However, in Fig. 7a we show pressure_head minus 0.76\*atmospheric_pressure, which is very similar and possibly better, we believe.*

387-391 these lines should be at the end of this section.
*Thank you, we moved these lines accordingly.*

427&428: The authors claim the increase of gas discharge as anomalous effect of different reasons except the anomalies as "probably merely accidental" at two sites (Soos and Bublák) during the summer 2016. These anomalies occurred a few weeks after the gas flow increase at Hartoušov due to the drilling process and the influence to the hydraulic regime. An assumption or specific interpretation should be added. Please think about the fluid interaction of the deeper horizons in the area (Cheb basin). For example, the gas eruption at the drill site H11 in the year 1957 induce an anomalous gas discharge and variations in the water levels in Františkovy Lázneˇ, about 2 km far. A reduced water table at the Hartoušov drill site could influence the hydraulic pressure regime in the nearby Cheb basin. This influence could trigger the ex-solution of $CO_2$ of the water table with a temporal delay at other locations (mofettes), similar to the atmospheric pressure effect.
*The eruption in 1957 and drilling in 2016 were rather different in scale. In 1957 the eruption lasted for several days with water fountain of about 50 m height compared to very small and short gas leakages during the 2016 drilling.*

464: Please take into account also that the strong drought period during the last summer reduces the level of the surrounding ground water table. This hydraulic pressure reduction induces an addition gas release as diffuse compo- nent and could reduce the total amount of gas discharge at the monitoring site.
*We newly checked this influence by comparing the records of gas flow and water level in three sites: Hartousov mofette, Bublak and Soos.*
*In the Hartousov mofette similar decay of water level and gas flow rate is visible until about February 2018. The later anticorrelation is probably caused by malfunction of the level logger (temperature dependent).*
*Bublak and Soos show water level minima in summer 2018 and 2019, which partially correlate with the lowered gas flow in summer. However, no clear relation is found for 2017.*
*Because of unclear relation we do not show this graph in the paper. We added few sentences commenting on this possible influence to the Discussion.*

[Figure]

477: because of missing evidences of this process, please add "... indicates the possible presence of..."
*We believe that the verb 'indicates' is a sufficient way to indicate that it is not fully proved (compared to verbs like proves, documents, etc...)*

Ad Technical comments

$CO_2$ – should be written with 2 subscript. The names of the ref- erences in the text should be outside the brackets, e.g. Fischer et al. (...), see line 68, 250, 259 a.s.o.
*Thank you, corrected*

439: considered as an 493 "discharges" is a better term here than "emanations"
*Thank you, corrected*

564 this reference is not mentioned in the text A few typesetting mistakes in the reference list
*Thank you, corrected*

The figure captions should be not in bold
*Thank you for this comment, this occurred by mistake, now it has bee corrected.*

---

## Author Comment (AC2) · 16 Apr 2020

Thank you for your helpful comments. We address all of them in our replies below and in most cases have modified the manuscript accordingly, which is in each case indicated in our reply.

Ad Comments

…For example, a statement about the performance of resistivity measurements, which were also used to estimate the gas bubble fraction although in a different set-up, would be interesting…
*We added few sentences to the end of Discussion commenting on the performance of the resistivity method.*

P2 L50: Could you please specify which fault zone is meant here?
*Thank you, this was not indicated. We clarify that this is related to the hypocenter trend.*

Table 1. CarbonNet monitoring network: Hartoušov, the 105.8 m deep borehole is not mentioned in the description of the network in the main text (unless I missed it). Is the borehole used as the reference borehole for the integral method described in the following?
*Thank you for this comment. No, this borehole is operated in a closed regime, so the integral method cannot be tested there. We added few sentences to clarify the origin of the borehole, in particular:*
*„In 2016, a 105.8 m deep borehole was drilled in the Hartoušov mofette with the aim of studying geo-bio interactions (Bussert et al., 2017). It showed a CO2 overpressure of 5 bars and was converted to a closed monitoring borehole with continuous measurements of downhole pressure and temperature and wellhead pressure. A broadband seismometer was installed in 70 m depth in the year 2019"*

P6 L183: "ceiling of the aquifer" the term seems strange to me, I assume it means simply the top of the aquifer.
*OK, we changed to top.*

P9 LL 260 – 264: ".... that the gas bubbles have to appear at the penetrated section of the Hartoušov borehole. This allows us to determine the mean volumetric fraction of the bubbles using eq. (3) with h1(t) = hm(t) being the hydraulic head measured at the depth dm = 4m, and h2(t) = he(t) being the hydraulic head measured in any depth below the bubble entry depth, which we suppose to be at the upper part of the penetrated section at de = 20.5m (Fig.3)."
The statement is confusing to me. If the gas bubbles enter the borehole at the penetrated section, how can the upper part of the penetrated section be below the bubble entry depth? Is the hydraulic head for he actually measured in the Hartoušov borehole or in the reference well mentioned before? I assume the latter is the case according to Figure 4. Could you please clarify the text here?
*You were right that the formulation was confusing. Our understanding is that the bubbles enter the borehole at the upper edge of the perforation, which was not clear before. Now we slightly changed the wording by modifying –'in any depth below the entry point' to 'at the bubble entry depth (or anywhere below)'*

P10 L 307: What is the observed mass flux at the teste site?
*Based on the measurements of Nickschick et al. (2015) (2-100 tons/day over an area of 350 000 m²) the flow rate through the borehole section would be in the order of 1E-8 kg/s. We added this estimate to the text.*

P 10: Section: Laboratory test of bubble fraction method: Could you please state clearly here which bubble fraction method was tested in the laboratory, the integral or the differential method? I assume the latter is the case.
*You are right, we mention it now in the text.*

P 11 LL 346 – 350 (Section: Laboratory test of bubble fraction method): I assume the statement, that the integral method performs better than the differential method, is purely based on the observed correlation of the field data and not supported by the laboratory tests.
*Yes, the reviewer is right because the integral method could not have been tested in the laboratory.*

Although, I agree with the statement it seems to be a bit out of context here.
Thank you for this comment. The section 2.5 in fact deals both with the laboratory tests and with the tests at Hartoušov. Accordingly, we renamed the section to 2.5 Tests of bubble fraction method

Furthermore, why is a different time window utilized for the differential method in Figure 5 and Figure 6?
*In Fig. 5 the aim was to find the volumetric fraction values with highest possible range so that we can compare the results with laboratory results – that's why time windows from 2016 (differential method) and 2018/2019 (integral method) were chosen. In case of Fig. 6, where we compare only integral and differential method itself, our aim was to choose the same time interval for both methods. Unluckily, due to many technical issues, we were only able to choose the plotted time windows (Jun-Sep 2018 and Jan-Apr 2019), which are close to each other and have approximately the same length.*

Maybe a separate section discussing the differences in more detail would be better here including the statement on P13 LL 432 – 436, which I assume refers back to Figure 6 and not Figure 5.
*Thank you for this comment. In fact, the discussion of Figs. 5 and 6 was a bit mixed up. Now we corrected the figure number from 6 to 5 and added a sentence introducing Fig. 6.*

P12 LL 385-386: This statement should be followed by paragraph P12 LL 392-399. The small rearrangement would make it easier for the reader to follow, that there is a large effect on the data due to barometric pressure variations and that these have to be corrected. Maybe that could also be explicitly mentioned, although it is implicitly clear.
*We have rearranged it accordingly, also in agreement with the Rev#1 recommendation.*

P8 L238 and 243: $\varphi_O$ should be capitalized
*Thank you, we have corrected it*

P10 LL 319-321: Figure 5 should be referenced here.
*Thank you, we have added it*

P11 L338: This should be Figure 5 and not Figure 6.
*Thank you, we have corrected it*